# *Perilla frutescens* Extracts Enhance DNA Repair Response in UVB Damaged HaCaT Cells

**DOI:** 10.3390/nu13041263

**Published:** 2021-04-12

**Authors:** Hyuna Lee, Eunmi Park

**Affiliations:** Department of Food and Nutrition, Hannam University, Daejeon 34054, Korea; hyuna916@naver.com

**Keywords:** *Perilla frutescens*, keratinocytes, cell proliferation, ultraviolet radiation, DNA repair

## Abstract

Physiological processes in skin are associated with exposure to UV light and are essential for skin maintenance and regeneration. Here, we investigated whether the leaf and callus extracts of *Perilla frutescens (Perilla)*, a well-known Asian herb, affect DNA damage response and repair in skin and keratinocytes exposed to Untraviolet B (UVB) light. First, we examined the protective effects of *Perilla* leaf extracts in UVB damaged mouse skin in vivo. Second, we cultured calluses using plant tissue culture technology, from *Perilla* leaf explant and then examined the effects of the leaf and callus extracts of *Perilla* on UVB exposed keratinocytes. HaCaT cells treated with leaf and callus *Perilla* extracts exhibited antioxidant activities, smaller DNA fragment tails, and enhanced colony formation after UVB exposure. Interestingly, keratinocytes treated with the leaf and callus extracts of *Perilla* showed G1/S cell cycle arrest, reduced protein levels of cyclin D1, Cyclin Dependent Kinase 6 (CDK6), and γH_2_AX, and enhanced levels of phosphorylated checkpoint kinase 1 (pCHK1) following UVB exposure. These observations suggest that the leaf and callus extracts of *Perilla* are candidate nutraceuticals for the prevention of keratinocyte aging.

## 1. Introduction

Skin is primarily composed of epidermis and dermis and exposed to a variety of environmental factors including UV radiation. UV exposure is a major factor of age-related changes such as pigmentary changes, thinning, wrinkling, and skin cancer development [1]. UV radiation is composed of Ultraviolet A (UVA), Ultraviolet B (UVB), and Ultraviolet C (UVC), which are defined by wavelength [2]. Exposure to UVB causes inflammation and immune changes that induce cellular senescence, direct DNA damage, and increases levels of oxidative stress, which can indirectly induce DNA mutagenesis. In response to DNA damage, the DNA response and repair pathway is activated in keratinocytes [3,4].

If not repaired properly, DNA damage in keratinocytes may result in the accumulation of mutations that prompt cell cycle arrest and promote skin aging. In particular, when exposed to UVB, phosphorylated checkpoint kinase 1 (pCHK1) is activated, this causes G1/S cycle arrest and reduces the proportion of cells in the S phase and skin repair response [5].

*Perilla frutescens (Perilla)* is an annual herb widely cultured in Asia. *Perilla* leaves are known to have antioxidant, anti-inflammatory, and anti-allergic effects and to be rich in polyphenols [6,7]. Calluses (as known as plant stem cells) are disorganized, non-differentiated plant cell masses and are induced using hormones, such as auxin or cytokinin from plant explants [8,9,10]. Calluses contain phenolic acid, which acts as an antioxidant, and other derivative sources. Recently, callus induction has been further developed to produce bioactive nutrient compounds [11,12].

Here, we investigated whether *Perilla* leaf extracts can induce DNA damage response and repair in UVB exposed mouse skin. In addition, we describe an efficient, rapid means of *Perilla* callus induction from *Perilla* leaves in different media in the presence of the growth regulators, 2,4-dichlorophenoxy acetic acid (2,4-D) kinetin, and organic additives. We then examined the effects of the leaf and callus extracts of *Perilla* on DNA damage response and repair in UVB exposed keratinocytes. This study shows that *Perilla* extracts are potential nutraceutical reagents for maintaining keratinocyte homeostasis after sun exposure.

## 2. Materials and Methods

### 2.1. UVB Irradiated Mouse Experiment

In the present study, Friend leukemia virus B (FVB) female mice (10 per group) were used as previously described [4]. After removing dorsal skin hair, *Perilla* leaf extracts of 5% *w*/*v* were applied in acetone to mouse skin. Controls were treated with acetone alone. Mice were then placed in separate compartments of a modified cage and 30 min after extract application, their backs were UVB irradiated. This process was performed 4 times/week for 2 weeks. The mount of UVB administered was progressively increased by increasing exposure times (i.e., 30, 60, 90, 120 s during the first week and 240, 270, 300, and 330 s during the second week (up to 100 mJ/cm^2^). Epidermal thickness was measured using photomicrographs (100 μM) of 5 sections per mouse from 10 mice. The software program used was Leica application suite version 4.8. All animal experiments were approved beforehand by the Institutional Animal Use and Care Committee (HNU 2016-6).

### 2.2. Preparation of Perilla Callus Induction

*Perilla frutescens* were collected from Geumsan, Chungcheongnamdo, South Korea. To induce callus growth, *Perilla* leaves were first disinfected in 5 mM salicylic acid solution for 2 min, 70% ethanol for 10 min, and 1% bleach (sodium hypochlorite) for 10 min, and then washed 3 times in sterile distilled water for 15 min. Sterilized leaves were cut into 1 × 1 cm^2^ pieces and inoculated into one of three media. For media preparation, agar 8 g/L, sucrose 10.09 g/L, and 3-(N-morpholino) propane sulfonic acid (MOPS) 0.5 g/L were added to each of; (1) Murashige & Skoog medium (M&S medium), (2) Murashige & Skoog modified medium (M&S modified medium), or (3) 2,4-dichlorophenoxy acetic acid medium (2,4-D medium). All media were adjusted to pH 5.7 and autoclaved at 121 °C for 15 min before use. Sterilized leaves were cultivated in media at room temperature for about 4 weeks. Calluses were then harvested and extracted with 70% ethanol. Extracts were then filtered, vacuum concentrated, and freeze-dried. The powdered extracts of calluses obtained using the three different media were used in the experiments detailed below.

### 2.3. ORAC (Oxygen Radical Absorbance Capacity) Assay

The peroxyl radical scavenging abilities of *Perilla* leaf and callus extracts were assessed using an ORAC assay [13]. Briefly, fluorescein (100 μL/80 nM) in 75 mM potassium phosphate buffer (pH 7.4) was added to triplicate wells in a black, flat-bottom, 96-well microplate. 2,2′-Azobis (2-amidino-propane) dihydrochloride (AAPH; 50 μL/80 mM), a peroxyl radical generator, was then added to the microplate, which was then immediately inserted into a SpectraMax i3x Platform (Molecular Devices, Lagerhausstrasse, San Jose, CA, USA). ORAC were measured every 2 min at 37 °C and expressed as Trolox equivalents (TE; μM). One ORAC unit was the equivalent to the net area provided by 1 µM Trolox.

### 2.4. Cell Culture and UV Irradiation

HaCaT cells (a human keratinocyte cell line) were a generous gift of Dr. Dae Joon Kim (University of Texas Health Rio Grande Valley) and cultured in Dulbecco’s modified Eagle’s medium containing 10% fetal bovine serum and 1% penicillin/streptomycin in a 5% CO_2_ atmosphere at 37 °C [14]. Samples were pretreated 12 h before UVB irradiation. Cells were washed twice with phosphate buffered saline (PBS) and irradiated with UVB using a UV source (Spectrolinker Xl-1000B UV crosslinker; New York, NY, USA) once at 30 mJ/cm^2^. After UVB irradiation, the medium was replaced with fresh serum-free medium. Cells were left for 3 h and then harvested for experiments.

### 2.5. Cell Viability Assay (MTT Assay)

The effects of *Perilla* leaf and callus extracts on HaCaT viability were determined using an 3-(4,5-Dimethylthiazol-2-yl)-2,5-Diphenyltetrazolium Bromide (MTT) assay [15]. Cells were plated at 5 × 10^5^ cells/well into 12-well plates, cultured for 24 h, and then treated with leaf and callus of *Perilla* for 12 h. MTT solution (5 mg/mL) was then added, and cells were incubated for 1 h. After removing media, formazan crystals were dissolved with DMSO and absorbances were measured at 570 nm.

### 2.6. Colony Formation Assay

Colony formation assays were used to measure cell viability by counting colonies formed by single cells [16]. HaCaT cells were treated for 12 h with *Perilla* leaf and callus extracts, irradiated with UVB, and then incubated in fresh Dulbecco’s Modified Eagle’s Medium (DMEM) medium at 37 °C in a 5% humidified CO_2_ atmosphere for 3 h. Cells were harvested with trypsin-Ethylenediaminetetraacetic acid (trypsin-EDTA), counted using a hemocytometer, and seeded in 6-well plates or 100 mm cell culture dishes. After incubation for 14 days, colonies were fixed with methanol, stained with crystal violet, counted manually, and photographed.

### 2.7. Comet Assay

Comet assays were used to assess DNA damage levels in cells [17]. HaCaT cells were treated with *Perilla* leaf or callus extracts for 12 h, exposed to UVB, and cultured for 3 or 12 h. Cells were then trypsinized, mixed with low melting agarose gel (LMA), and dispersed onto precoated slides with normal melting agarose (NMA). Cells were then covered with cover glasses and stored at 4 °C, and cover glasses were removed after the gel had hardened. Slides were soaked in pre-chilled alkali lysis buffer (2.5 M NaCl, 100 mM Na_2_EDTA, 10 mM Tris, 1% N-lauryl-sarcosinate, pH 10; 1% Triton-X-100 and 10% DMSO) at 4 °C for 60 min in the dark, and then in pre-chilled electrophoresis buffer (300 mM NaOH, 100 mM Na_2_ EDTA, pH > 13) at 4 °C for 40 min in the dark. Slides were then placed in a horizontal electrophoresis chamber filled with cold electrophoresis buffer, electrophoresed at 25 V and 300 mA for 20 min, washed with tris buffer (pH 7.4), dried, stained with ethidium bromide (20 μL/mL), and observed under a fluorescence microscope (Leica, Wetzlar, Germany). Images were taken using a CCD camera (Nikon, Tokyo, Japan) and analyzed using a Comet image analysis program (Kinetic image 4.0, Caliper Life Sciences, Alameda, CA, USA). Degrees of DNA damage in HaCaT cells were quantified using DNA fragment percentages as DNA tails.

### 2.8. Fluorescence-Activated Cell Sorting (FACS) Analysis

Cell cycle distributions were determined by propidium iodide staining [18]. HaCaT cells were cultured in 6-well plates, treated with *Perilla* leaf or callus extracts (0.1 μg/mL) for 12 h, UV irradiated, and incubated in fresh DMEM medium at 37 °C for 3 h. Harvested cells were immediately fixed in 70% ethanol at 4 °C overnight, washed once with PBS, stained with propidium iodide/RNase staining solution (Invitrogen, Waltham, MA, USA), and analyzed by flow cytometry (Beckman Coulter, Brea, CA, USA). The percentages of cells in phases of the cell cycle were calculated using CytExpert 1.0 software (Beckman Coulter, Brea, CA, USA).

### 2.9. Western Blot Analysis

HaCaT cells were cultured in 6-well plates, treated with *Perilla* leaf or callus extracts for 12 h, UVB-irradiated, and then incubated for 3 h at 37 °C [14]. Following incubation, cells were harvested with trypsin-EDTA, resuspended in lysis buffer on ice for 40 min, centrifuged at 13,714× *g* for 20 min at 4 °C, and protein concentrations in supernatants were determined using a NanoDrop (NanoDrop Lite spectrophotometer, Waltham, MA, USA). Equal amounts of proteins were subjected to SDS-PAGE and then transferred to nitrocellulose membranes, which were reacted with primary antibodies; pCHK1 (Serine 345), γH2AX (both from Cell Signaling, Danvers, MA, USA), CyclinD1, Cell division protein kinase 6 (CDK6), and β-actin (all antibodies from Santa Cruz Bicycles, Santa Cruz, CA, USA) at 4 °C overnight. The following day, membranes were incubated with anti-mouse or anti-rabbit IgG horseradish peroxidase-conjugated secondary antibodies for 40 min at room temperature. Blots were detected using the West Pico chemiluminescent kit (Thermo, Rockford, IL, USA) and visualized using LAS 4000 chemiluminescent image analyzer (Fuji, Tokyo, Japan).

### 2.10. Statistics

Results were analyzed by ANOVA with Tukey’s test. The analysis was conducted using SPSS-PC ver. 23.0 (SPSS Inc., Chicago, IL, USA). For the mouse experiments, results are presented as the means ± standard errors (SEs) of 10 mice per group. Other results are presented as means ± SEs or SDs of three independent experiments, as indicated. Data shown are representative of three independent experiments. Statistical significance was accepted for *p* values < 0.05.

## 3. Results

### 3.1. The Effect of Perilla Leaf Extracts on Mouse Skin Exposed to UVB

To determine the effects of the extracts on UV-induced skin damage, the dorsal skins of FVB mice were shaved and treated with *Perilla* leaf, callus extracts (5% *w*/*v* in acetone), or acetone (controls) before UVB exposure once daily for two weeks. Two weeks after UVB treatment, mice were sacrificed and skin tissue sections were fixed for hematoxylin and eosin (H&E) staining. Epidermal thickness was measured by microscopy. Acetone treatment (the control group) increased epidermal thickness after 2 weeks of treatment (Figure 1A,B). Interestingly, *Perilla* leaf extract reduced UVB-induced epidermal thickness increases (*p* < 0.001). Neither acetone nor *Perilla* leaf extract influenced epidermis thickness when the skin was not UVB exposed.

### 3.2. Induction of Perilla Callus Formation Using Different Media

Next, we established a means of culturing *Perilla* calluses, a well-known nutraceutical [12]. Calluses were induced from *Perilla* leaves over four weeks using three media types: M&S medium, M&S modified medium, and 2,4-D medium. Interestingly, *Perilla* callus induction was observed after five days of culture on 2,4-D medium and proliferated after nine days (Figure 2A,B). In contrast, induction of *Perilla* calluses was observed after nine days of culture on M&S modified medium, but *Perilla* calluses barely proliferated. Callus formation was more rapid in 2,4-D medium than in the other two media (*p* < 0.001, Figure 2A). *Perilla* calluses produced on 2,4-D medium, M&S modified medium, or M&S medium in five weeks were weighed (Figure 2C), and the amount of callus produced was larger on 2,4-D medium than on M&S modified medium (20.38 mg versus 5.78 mg, respectively). Callus formation was not induced in M&S medium. These observations showed 2,4-D medium containing 2,4-dichlorophenoxy acetic acid (2,4-D) provided an effective medium for *Perilla* callus induction and suggested that 2,4-D may be crucial for *Perilla* callus induction.

### 3.3. Perilla Leaf and Callus Extracts Had Antioxidant Effects and Promoted DNA Repair in UVB Exposed Keratinocytes

The antioxidative activities of *Perilla* leaf and callus extracts were assessed using ORAC assays using ascorbic acid as an antioxidant. We found the antioxidant activities of *Perilla* leaf extract was greater than those of callus extracts produced using 2,4-D medium or M&S modified medium (see Figure 3A). Interestingly, the antioxidant activities of *Perilla* callus extract induced using 2,4-D medium were greater than those induced using M&S modified medium (*p* < 0.005, Figure 3A). Therefore, we examined the effects of *Perilla* callus extract from calluses induced using 2,4-D medium and *Perilla* leaf extract in subsequent studies.

The effects of the *Perilla* leaf and callus extracts on UVB-induced DNA damage were investigated using human keratinocytes (HaCaT cells). In Comet assays, the tail region containing DNA fragments provides an index of DNA damage. We observed that UVB (50 mJ/cm^2^) treatment induced longer DNA tails than UVB untreatment in keratinocytes (27.5% versus 6.6 % for UVB+ versus UVB- in the DMSO treatment, as a negative control group; *p* < 0.001, Figure 3B). Moreover, we found that HaCaT cells treated with *Perilla* leaf or callus extract (0.1 μg/mL) and then UVB, had smaller Comet tails than cells treated with DMSO and then UVB (*p* < 0.001, Figure 3B). We also treated HaCaT cells with caffeic acid, a derivative of cinnamic acid and a bioactive constituent of *Perilla* leaves, as a positive control. We found that *Perilla* leaf, callus extract (0.1 μg/mL), or caffeic acid (150 μM) reduced DNA tails after UVB exposure (Figure 3B, *p* < 0.001; Appendix A). Our findings suggest that both *Perilla* leaf and callus extracts have potential use for repairing UV-induced DNA damage.

### 3.4. Perilla Leaf and Callus Extracts Enhanced UVB Irradiated Keratinocyte Survival

First, we measured HaCaT cell viabilities after exposure to different doses of *Perilla* leaf and callus extracts using an MTT assay. HaCaT cells were treated with extracts at concentrations of 0.01, 0.1, 1, 10, or 100 μg/mL (see Figure 4A). We found 85.8% of keratinocytes remained viable after treatment with *Perilla* leaf extract at 10 μg/mL, and that 90% and 81.4% of keratinocytes remained viable after treatment with *Perilla* callus extract at 10 or 100 μg/mL, respectively (Figure 4A). The observations suggest that *Perilla* leaf and callus extracts had no toxic effects on keratinocytes.

Then, colony formation assays were performed to determine whether *Perilla* leaf and callus extracts influence cellular survival after exposure to UVB. DMSO treatment post UVB irradiation (controls) significantly reduced HaCaT cell colony formation (99% and 60% for UVB untreated and UVB treated cells, *p* < 0.05, Figure 4B,C). However, *Perilla* leaf and callus extracts inhibited UVB-induced reductions in colony formation (*p* < 0.05, Figure 4C).

### 3.5. Perilla Leaf and Callus Extracts Enhanced DNA Repair Signaling and Protected HaCaT cells from UVB-Induced Cell Cycle Changes

*Perilla* leaf and callus extracts enhanced the levels of pCHK1(S345) protein (an inducer of DNA repair signaling) more than DMSO treatment in UVB exposed HaCaT cells (Figure 5A,B; Appendix A). Moreover, *Perilla* callus extracts reduced levels of γH2AX protein (a hallmark of DNA damage) more than *Perilla* leaf extract in UVB exposed keratinocytes (Figure 5B). These data suggest that *Perilla* callus extracts reduce DNA damage after UVB irradiation in keratinocytes.

Increased pCHK1(S345) protein expression affects the cell cycle. Therefore, we examined cell cycle profiles to investigate the protective effects of *Perilla* leaf and callus extracts on UVB irradiated keratinocytes.

Synchronous culture with chemical agents are in the same growth stage and provide less variation data over non-synchronous cells for studying cellular levels [19]. However, the synchronized drugs used resulted in side effects. We chose an asynchronized method, and its analysis of changes (Δ) in the cell cycle involved less perturbation of biological systems and drug interaction between chemical reagents and *Perilla* extracts. HaCaT cells were treated with *Perilla* leaf or callus extract overnight, irradiated with UVB (30 mJ/cm^2^), and 3hrs later, PI staining and flow cytometry analysis were used to determine cell cycle profiles (Figure 5C,D).

UVB increased the percentage of HaCaT cells in the G0/G1 phase (from 50.5% to 60.9%), indicating that it inhibited cell growth by causing G1/S arrest. We observed that the percentage of cells in the G0/G1 phase after UVB exposure was increased by *Perilla* leaf and callus extracts (Figure 5D, *p* < 0.05). Furthermore, 0.1 μg/mL of *Perilla* callus extract increased the change of proportion of HaCaT cells in the G0/G1 phase, as compared with DMSO treatment after UV treatment (Figure 5D, 10.4 ± 0.8 versus 19.1 ± 1.3 of change (Δ) of DMSO treatment versus change (Δ) of 0.1 μg/mL callus extract, *p* < 0.05).

We also found that 0.1 μg/mL of *Perilla* callus extract decreased the proportion of cells in the S phase as compared with DMSO treatment after UV exposure (Figure 5D, −7.4 ± 0.9 versus −12.9 ± 1.1 of change (Δ) of DMSO treatment versus change (Δ) of 0.1 μg/mL callus extract, *p* < 0.05). Our observations suggest that *Perilla* extract, especially callus, arrested UVB-induced cell cycle phase at G1/S in HaCaT cells.

### 3.6. Perilla Leaf and Callus Extracts Regulated G1/S Cell Cycle Genes in UVB Exposed HaCaT Cells

At the G1 phase of the cell cycle, the cyclin D1 and CDK6 genes are directly involved in cell cycle regulation. Moreover, G1/S cell cycle arrest is associated with reductions in cyclin D1 and CDK6 protein levels after UVB exposure [19]. We found that cyclin D1 and CDK6 protein expressions were reduced in HaCaT cells irradiated with 30 mJ/cm^2^ of UVB (Figure 6), and that *Perilla* leaf and callus extracts reduced significantly (Figure 6, *p* < 0.05). These results suggest that *Perilla* extracts may cause G1/S cell cycle arrest after UVB exposure by reducing cyclin D1 and CDK6 protein levels.

## 4. Discussion

It has been reported that UV-induced damage increases the levels of pCHK1protein expression during DNA damage and repair response [5,20]. In this study, we examined the effects of *Perilla* leaf and callus extracts in UVB exposed keratinocytes.

Previous studies have investigated the roles played by ubiquitin-specific protease 1 (USP1) in DNA damage response and on cellular functions and genetic instability in keratinocytes [14]. Activation of the DNA response and repair pathway induces cell proliferation and protects against senescence, and UVB radiation-mediated DNA damage reduces colony-formation potential and cell viability and destabilizes DNA in keratinocytes. Our results show that UVB significantly induced DNA breaks and reduced colony formation by HaCaT cells, whereas *Perilla* leaf and callus extracts both suppressed these UVB induced effects. Persistent exposure to UVB also increases DNA lesions, fragments, and mutations, and leads to premature skin aging, and interestingly, the present study shows *Perilla* leaf extract protects against DNA damage and enhanced cell proliferation in UVB exposed keratinocytes.

Links between UVB exposure, skin cell proliferation, and aging can be observed when the DNA repair and response pathway becomes dysfunctional. Defects in the DNA repair and response pathway enhance genomic instability in keratinocytes during aging [1]. Furthermore, the CHK1 gene activates the DNA repair systems that repair UVB-induced DNA lesions [20]. We found *Perilla* leaf and callus extracts enhanced pCHK1 protein expression and reduced γH2AX protein expression in UVB exposed keratinocytes. Accordingly, our findings suggest that both extracts increase keratinocyte proliferation by upregulating pCHK1 protein levels.

UV-induced reactive oxygen species (ROS) are responsible for DNA damage in skin. ROS are produced endogenously by metabolic processes and exogenously by the effect of UV on skin. Moreover, persistently elevated ROS and oxidative DNA damage levels are associated with poorer colony-forming ability [21]. In the present study, *Perilla* leaf and callus extracts effectively reduced ROS levels, increased DNA damage repair, and keratinocyte colony-forming ability, and were also found to have antioxidant effects. Thus, *Perilla* extracts may be a useful modality for the prevention of UV-induced keratinocyte aging.

DNA damage simultaneously triggers the cell cycle checkpoint and activates the DNA repair pathway [20]. Checkpoint proteins such as cyclin A, D, E, and K are activated by UV exposure and elicit cell cycle arrest by G1/S or G2/M signaling and facilitate DNA repair [19]. We found *Perilla* callus extracts induced G1/S cell cycle arrest by reducing cyclin D1 and CDK6 levels, and that both extracts, especially callus extract, effectively regulated cell cycle checkpoint proteins and triggered DNA repair response in UVB exposed keratinocytes. Thus, our findings indicate that *Perilla* extracts may effectively regulate the cell cycle and the G1/S cell cycle checkpoint.

Interestingly *Perilla* extracts also increased keratinocyte proliferation, which suggests that they might protect against keratinocyte aging. The antioxidant effects of *Perilla* leaf extracts have been previously studied in pharmacological properties [7]. *Perilla* leaf has been used as a traditional medicine to treat depression, anxiety, tumors, coughs, allergies, and other conditions, and our findings suggest that *Perilla* leaf and callus extracts could be used to treat keratinocyte damage and UV exposure-induced keratinocyte conditions [6,22]. *Perilla* extracts have been reported to contain the following active compounds: phenolic acids, flavonoids, anthocyanins, volatile compounds, triterpenes, phytosterols, fatty acid tocopherols, policosanols, rosmarinic acid, luteolin, and tormentic acid [22]. Although our preclinical data have yielded promising results, further studies are needed to determine the effects of active compounds in *Perilla* extracts.

2,4-D is a well-known plant growth regulator and an analog of auxin (a plant growth hormone). It has been reported that callus induction and lycopene production by the leaves of *Barringtonia racemose* were dependent on 2,4-D concentration in media [23]. Our study was conducted to establish a means of culturing calluses and regenerating *Perilla* plants using different plant hormones. *Perilla* leaf extract exhibited a better callus response when 2,4-D medium was supplemented with 2,4-dichlorophenoxy acetic acid (2,4-D), than when it was supplemented with kinetin, and a maximum callus score of 4.28 was recorded when leaves were cultured on MS modified media containing naphthaleneacetic acid. Furthermore, given increasing demands for natural bioactive products, plant cell cultures using plant growth regulators offer an attractive means of overcoming the limitations of extraction-based methods for obtaining biomaterials for nutraceutical studies.

It should be noted the amount of callus produced in the present study was relatively small and not enough to examine its effects on skin in vivo. Nevertheless, both *Perilla* leaf and callus extracts were found to have potent antioxidant effects and to increase DNA repair response in keratinocytes, and thus, to offer potential means of preventing and treating keratinocyte photoaging. Finally, *Perilla* extract, leaf or callus, presents better potential as source of keratocyte antiaging products.

## Figures and Tables

**Figure 1 nutrients-13-01263-f001:**
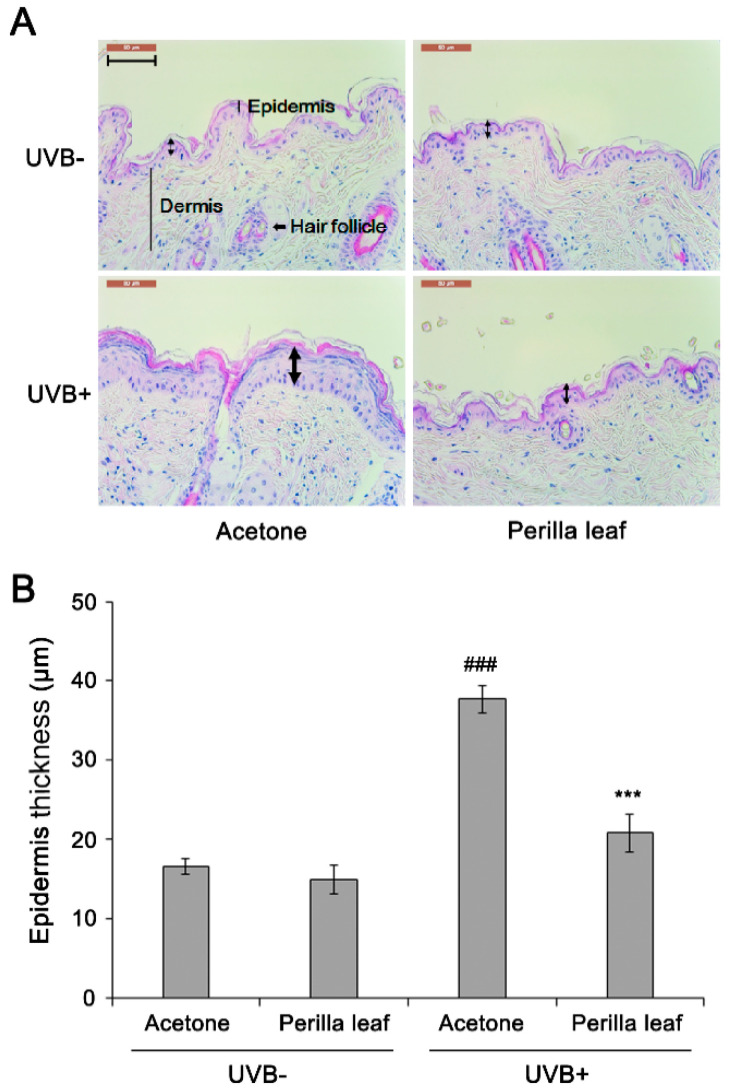
*Perilla* leaf extract suppressed UVB-induced epidermal thickening in FVB mice. (**A**) H&E-stained section of mouse dorsal skin after Ultraviolet B (UVB) exposure. *Perilla* leaf extract (5% *w*/*v* in acetone) was applied to mouse skin, which was then irradiated with UVB for 30 min. UVB-induced epithelial cell proliferation was evaluated in each indicated group (ten mice per group), after 2 weeks of UVB irradiation was completed. The photograph shows skin from a representative mouse. Bold arrows indicate epidermis. The scale bar represents 100 μm. (**B**) Quantification of epidermis thickness in UVB-induced epithelial layers. Data are expressed as means ± SDs. ### *p* < 0.001 as compared with non-radiated mice by the t-test. *** *p* < 0.001 as compared with the UVB-irradiated control group by the *t*-test.

**Figure 2 nutrients-13-01263-f002:**
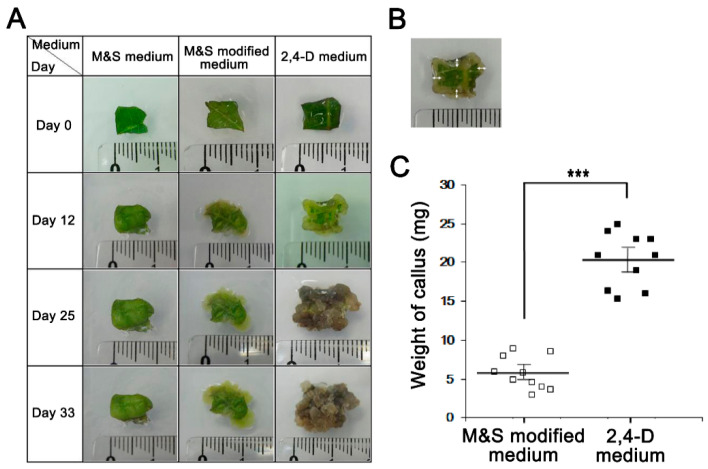
**The effect of *Perilla* callus formation in 2,4-D and two M&S media.** (**A**) *Perilla* callus induction dependence on the three medium types. Representative photographs showing *Perilla* calluses on culture days 0 to 33. (**B**) Calluses that formed from the four corners of *Perilla* leaves were measured using a ruler. (**C**) Weights of calluses by medium type. We measured the biomasses of *Perilla* calluses after culture for 5 weeks. Dots and bars represent the means and SDs of three independent samples. *** *p* < 0.001 as determined by the t-test.

**Figure 3 nutrients-13-01263-f003:**
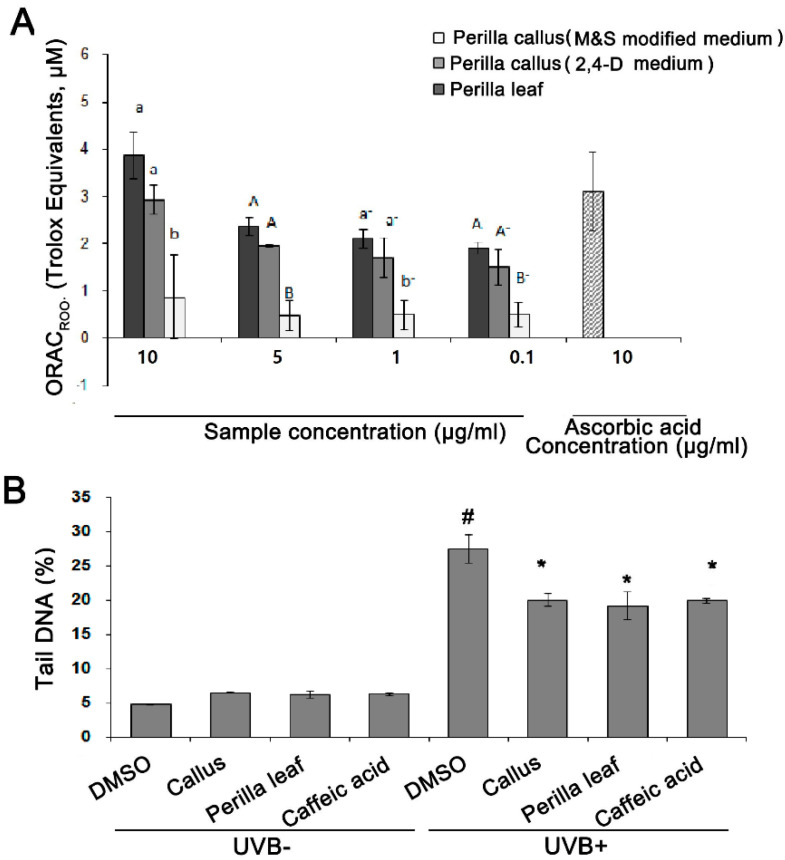
*Perilla* leaf and callus extracts exhibited antioxidant effects and enhanced DNA repair as determined by the Comet assay in UVB exposed HaCaT cells. (**A**) Antioxidant activities of *Perilla* leaf and callus extracts as determined by the oxygen radical absorbance capacity (ORAC)_ROO·_ assay in HaCaT cells. The results presented are the means ± SDs of triplicate determinations. Data shown are representative of three independent experiments. Values were compared with those obtained for *Perilla* callus extracts induced by MS modified medium and 2,4-dichlorophenoxy acetic acid (2,4-D) medium, and leaf extracts at the same concentrations, as 1 uM trolox equivalent. *p* < 0.01 as determined by Duncan’s test. (**B**) DNA damage was assessed using the Comet assay. Tail DNA percentages were determined. Comet images revealed different degrees of DNA damage. HaCaT cells were cultured in 6-well plates and pretreated with *Perilla* leaf or callus extracts (at 0.1 μg/mL), caffeic acid (150 μM; the positive control), or DMSO (the negative control) for 12 h, exposed to UVB (50 mJ/cm^2^), and then cultured for 12 h later. The bar graph was calculated from three independent experiments. Images of fluorescence intensities were obtained using the Comet assay. # *p* < 0.05; significant versus UVB non-treated and DMSO controls. * *p* < 0.05; significant versus UVB treated and DMSO controls. Different letters indicate significant intergroup differences (*p* < 0.01).

**Figure 4 nutrients-13-01263-f004:**
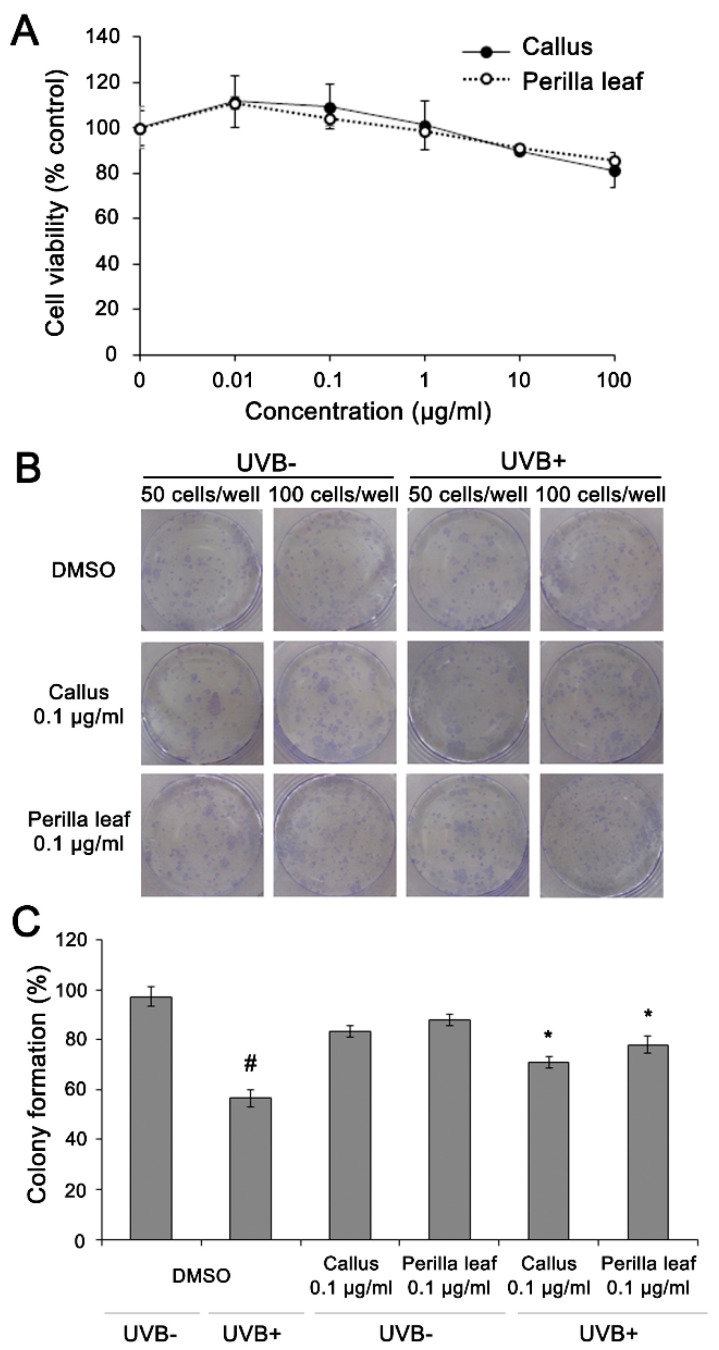
***Perilla* leaf and callus extracts prevented the suppression of HaCaT proliferation by UVB.** (**A**) 3-(4,5-Dimethylthiazol-2-yl)-2,5-Diphenyltetrazolium Bromide (MTT) assay of HaCaT cells treated with *Perilla* leaf and callus extracts. Results are expressed as means ± SDs. Different letters indicate significantly different (*p* < 0.05) by Duncan’s test. (**B**,**C**) HaCaT cells were treated with *Perilla* leaf or callus extracts (0.1 μg/mL) overnight, irradiated with UVB (30 mJ/cm^2^), harvested, and reseeded at 50 or 100 cells/well in 6 well plates. Colony formation was observed 14 days after seeding. (**B**) Representative colony-forming assay images after methanol fixation and crystal blue staining. (**C**) Colony numbers were counted and recorded. Colony formation rates of HaCaT cells (determined by analyzing 50 cells/well). Results are presented as the means ± SDs of three plates. # *p* < 0.05 versus UVB non-treated and DMSO controls. * *p* < 0.05 versus the UVB treated and DMSO controls.

**Figure 5 nutrients-13-01263-f005:**
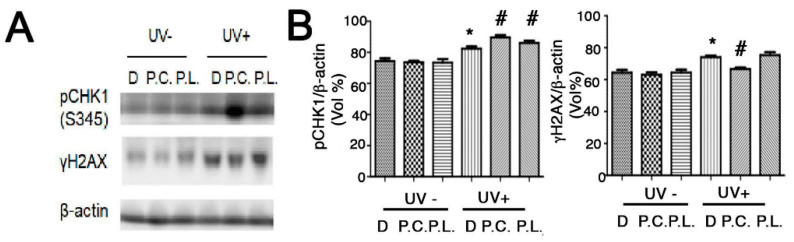
**Effect of *Perilla* leaf and callus extracts on the cell cycle distribution of UVB exposed HaCaT cells.** (**A**) Representative Western blot for phosphorylated checkpoint kinase 1 (pCHK1) (S345) and γH2AX proteins in HaCaT cells treated with *Perilla* leaf or callus extracts at 0.1 μg/mL (D: DMSO treatment; P.C; *Perilla* callus; P.L: *Perilla* leaf). β-Actin was used as the loading control. Proteins were analyzed using the Fusion image system (Fuji, Japan). Cells were treated with *Perilla* extracts overnight, washed with Phosphate-buffered saline (PBS), and irradiated with UVB (30 mJ/cm^2^), and 3 h later, pellets were harvested for Western blotting. (**B**) Results are presented as the means ± SDs of three biological replicates in two independent measurements (please see all images in supplementary Appendix A). * *p* < 0.05 versus UVB non-treated controls. # *p* < 0.05; versus UVB treated comparisons. (**C**,**D**) Cell cycle analysis was performed by propidium iodide staining in asynchronized HaCaT cells. The proportions of cells in different phases of the cell cycle were determined by flow cytometry. (**C**) Representative histograms show cell cycle distributions. HaCaT cells were subjected to fluorescence-activated cell sorting (FACS) analysis. Histograms show side scatter (SS) and forward scatter (FS) for all events. The parameters of FACS analysis are referred to as FSC-A: Forward Scatter-Area; FSC-H: Forward Scatter-Height and SSC-A: Side Scatter-A. (D) Bar chart presentations of cell cycle distributions after treatment with *Perilla* leaf or callus extract and UVB (30 mJ/cm^2^). The cell cycle distribution table shows percentages of cells in the three phases (G0/G1, S, and G2/M). Results are presented as the means ± SDs of triplicate plates per sample. * *p* < 0.05; significant versus UVB non-treated controls. + *p* < 0.05; significant versus UVB non-treated controls on G0/G1 phase. Δ; change of UVB treated from UVB untreated in each cell cycle phase. # *p* < 0.05; significant versus change (Δ) of DMSO treatment.

**Figure 6 nutrients-13-01263-f006:**
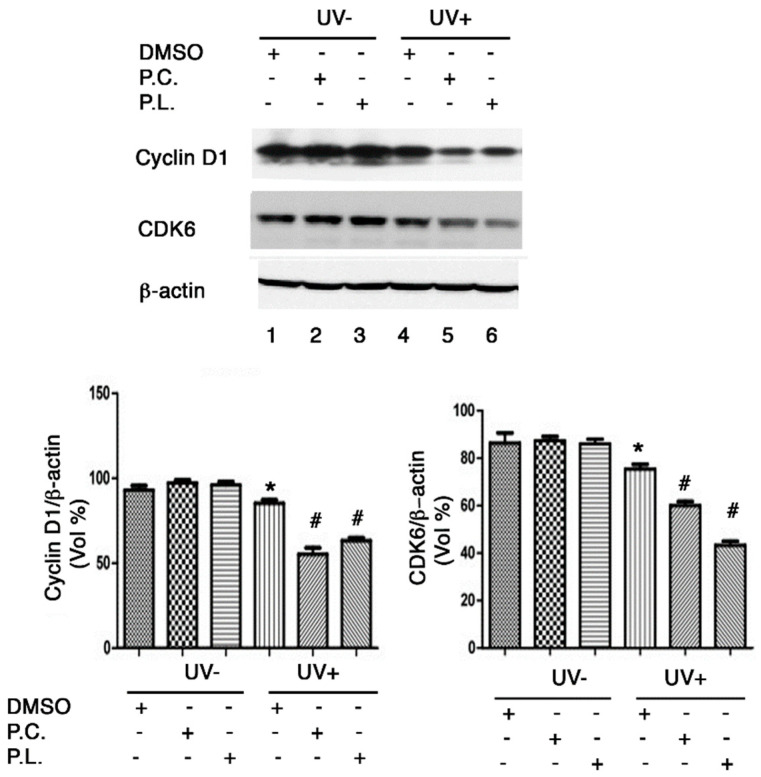
Western blot analysis of G1/S cell cycle regulatory protein expressions in HaCaT cells treated with *Perilla* leaf or callus extracts. Effects of *Perilla* leaf and callus extracts on the UVB-induced protein levels of Cyclin D1 and CDK6 in HaCaT cells. Cells were treated with *Perilla* leaf, callus extracts (0.1 μg/mL), or DMSO (negative control) overnight, washed with PBS, irradiated with UVB (30 mJ/cm^2^), and harvested 3 h later. Pellets were subjected to Western blotting using β-actin as a loading control. (Top panel) Representative Western blot for cyclin D1 and CDK6 proteins in HaCaT cells treated with Perilla leaf or callus extracts at 0.1 μg/mL (D: DMSO treatment; P.C; Perilla callus; P.L: Perilla leaf). (Bottom panel) Proteins were analyzed using the Fusion image system (Fuji, Japan). Data are representative of three independent experiments and expressed as means ± SDs. * *p* < 0.05; significant versus UVB non-treated controls. # *p* < 0.05; significant versus UVB treated controls.

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
