# Peer review of "Perilla frutescens* Extracts Enhance DNA Repair Response in UVB Damaged HaCaT Cells"

_nutrients, 2021, doi:10.3390/nu13041263_

Round 1

Reviewer 1 Report

Manuscript of Lee and Park describe the preparation of a plant extract from Perilla frutescens, and its effects in preventing DNA damage induced by UVB exposure. The study is well designed, the introduction provides sufficient background, the manuscript is well organized, and I believe it is interesting for the readers of Nutrients. 

It was significantly improved, however, some details must be highlighted:

  • There are still some parts of the manuscript where the name of the species is not in italic.
  • line 75 - I believe that lyophilized and freeze-dried are synonyms. Please confirm.
  • line 149 - the name of the test is Tukey's test, not Turkey's.
  • line 341-342 - "These observed effects of Perilla extract on DNA repair could help explain the significance  of ROS-induced DNA damage during skin aging." Please explain how the effects observed  explain the significance of ROS-induced DNA damage during skin aging.
  • line 356-359 - The sentence would be better if written like this:  "Perilla extracts have been reported to contain the following active compounds: phenolic acids, flavonoids, anthocyanins, volatile compounds, triterpenes, phytosterols, fatty acid tocopherols, policosanols, rosmarinic acid, luteolin, and tormentic acid [22]."
  • It should be more clear in the conclusions and in the abstract, if the authors added a sentence discussing which extract, leaf or caulus, presents better potential as source of skin antiaging products.

Reviewer 2 Report

My original comment#3:

Fig. 3B, C: From images of Fig.3B, tail DNA was not clearly observed even in UVB-irradiated cells with DMSO treatment. The reviewer also thinks that 17% of tail DNA even after UVB irradiation (Fig.3C) is too low to determine the protective effects of perilla extracts against DNA damage. To examine the protective role of perilla extracts against UVB-induced comet formation, cells should be irradiated with higher dose of UVB and effects of perilla extracts should be re-examined.

Author’s Response:

Yes, we agreed it was possible to lower percentage of DNA damage in DMSO treatment after UVB irradiation. However, UVB of above 30 mJ/cm2 was a higher dose in vitro assay as previous study (Mol Cell, Park et al 2014). Indeed, we found that perilla extracts reduced tail DNA (%) compared to UVB irradiation ( 8.5% versus 17% of perilla extract versus DMSO treatment in UVB treatment, *P<0.05) in new figure 3B.

As I said in my original comment, only 17% of tail DNA even after UVB irradiation is too low to determine the protective effects of perilla extracts against DNA damage. If the authors cannot show a clearer difference with their comet assay, then other methods such as the TUNEL assay should be performed to support their conclusion.

My original comment#4:

Figs. 5A, 6B: In Fig.5A, downregulation of gH2AX by perilla callus extracts is subtle. In Fig. 6, downregulation of cyclinD1 and CDK6 by 1 ug/mL perilla leaf extracts seems not to be reliable. Authors should perform quantification of western blot data and statistical analysis to determine whether there are differences with statistical significance.

Author’s Response:

We provided quantification data of western blot data in the new figure 5B and Figure 6 as reviewer requested.

Fig.5B: The difference is so small that it is not appropriate to draw any conclusions from only duplicate experiments. Authors did not mention whether they performed technical replicates or biological replicates. Authors should perform at minimum three biological replicates and show the image of all western blot data as supplementary figure or for reviewer only data.

My original comment#6:

Fig. 5D, E: Authors insisted that 0.1 ug/mL perilla callus extracts increased proportion of cells in G0/G1 phase (line 261) and S phase (line 279), as compared with DMSO treatment after UVB irradiation. However, the percentages of UVB and DMSO-treated cells in G0/G1 (60.9%) and S phase (26.0%) were similar to those of 0.1 ug/mL callus extracts-treated cells (percentages of G0/G1 and S phase cells were 61.9% and 26.2 %, respectively).

Author’s Response:

We provided new sentences and new figure 5D table and legend. Also, The methods for UVB irradiation of change (delta) calculating is now described in the figure legend of Figure 5D; 0.1 μg/ml of Perilla callus extract increased the change of proportion of HaCaT cells in the G0/G1 phase, as compared with DMSO treatment after UV treatment (Figure 5D, 10.4±0.8 versus 19.1±1.3 of change (Δ) of DMSO treatment versus change (Δ) of 0.1ug/ml callus extract, P<0.05).

In my opinion, if the population of cells in G0/G1 or S phase was not significantly different between DMSO-treated cells and callus extracts-treated cells under UVB-irradiated condition, the change of cell population between non-treated or UVB-irradiated sample does not mean anything and does not support the conclusion that callus extracts arrest cell cycle at G1/S phase after UVB irradiation.

My original comment#8:

Authors described “perilla callus extracts induced G1/S cell cycle arrest by reducing CDK6 levels, and that perilla extracts, particularly callus extracts, effectively regulate cell cycle-check point proteins and trigger DNA repair response in UVB exposed keratinocytes” (line 334). Authors should examine the effect of perilla leaf extracts on cell cycle arrest, levels of CDK6 and cell cycle check point proteins and DNA repair in mouse skin after UVB exposure. The reviewer suggests that the analysis would be performed at earlier time points than 2weeks after UVB exposure.

Author’s Response:

We agree that performance of earlier time points after UVB exposure might have strengthen our study. Still, our results clearly show that perilla extracts induced G1/S cell cycle arrests in 30 mJ/sec of UVB irritated keratinocytes, consistent with reduced CDK6 protein levels. Also, we have additional evidence that cyclin D1 and CDK6 protein expressions were reduced in keratinocytes irradiated with 30 mJ/cm2 of UVB (new Figure 6A).

Since the conclusion of the manuscript is that Perilla frutescens extracts enhances DNA repair response in UVB damaged “skin” as title of paper, authors must show DNA repair in not only HaCaT cells but mouse “skin” after UVB exposure. This experiment is absolutely required to justify of the conclusion of this study.

Round 2

Reviewer 2 Report

Now the manuscript was improved by additional experiments.

This manuscript is a resubmission of an earlier submission. The following is a list of the peer review reports and author responses from that submission.

Round 1

Reviewer 1 Report

In this manuscript, Lee and Park showed that leaf of perilla extracts played protective role against UVB-induced epidermal thickening in mouse skin. They also showed perilla extracts showed antioxidant activity, reduced DNA damage, and enhanced colony formation in HaCaT cells after UVB exposure. Additionally, they showed the perilla extracts changed the amount of pCHK, gH2AX, Cyclin D, and CDK6 in UVB-treated HaCaT cells. While the topic of this manuscript is of general interest, the quality of the manuscript is rather low and the amount of data presented is sparse. In its current state results are overinterpreted. Therefore, I suggest further experiments, which would support their conclusion and improve the quality of the manuscript substantially.

  1. Fig. 1 A: How was the epidermal thickness measured? Please indicate in the text as well as in the materials and methods part.
  2. Fig. 3A: Since effects of perilla extracts on DNA damage (Fig. 3B, C), colony formation (Fig.4B, C), cell cycle (Fig.5) were examined at a dose of 0.1 or 1 ug/mL, authors should examine whether 0.1 and 1 ug/mL of perilla extracts show the antioxidative activity.
  3. Fig. 3B, C: From images of Fig.3B, tail DNA was not clearly observed even in UVB-irradiated cells with DMSO treatment. The reviewer also thinks that 17% of tail DNA even after UVB irradiation (Fig.3C) is too low to determine the protective effects of perilla extracts against DNA damage. To examine the protective role of perilla extracts against UVB-induced comet formation, cells should be irradiated with higher dose of UVB and effects of perilla extracts should be re-examined.
  4. Figs. 5A, 6B: In Fig.5A, downregulation of gH2AX by perilla callus extracts is subtle. In Fig. 6, downregulation of cyclinD1 and CDK6 by 1 ug/mL perilla leaf extracts seems not to be reliable. Authors should perform quantification of western blot data and statistical analysis to determine whether there are differences with statistical significance.
  5. Fig. 5B: What did the panels of FSC/SSC profiles show? Please include the explanation of the data in the text as well as the figure legend.
  6. Fig. 5D, E: Authors insisted that 0.1 ug/mL perilla callus extracts increased proportion of cells in G0/G1 phase (line 261) and S phase (line 279), as compared with DMSO treatment after UVB irradiation. However, the percentages of UVB and DMSO-treated cells in G0/G1 (60.9%) and S phase (26.0%) were similar to those of 0.1 ug/mL callus extracts-treated cells (percentages of G0/G1 and S phase cells were 61.9% and 26.2 %, respectively).
  7. Authors described that the leaf and callus of perilla enhanced pCHK1 protein expression and reduced gH2AX protein expression in UVB exposed keratinocytes (line 319). However, leaf extracts neither enhanced pCHK1 level nor reduced gH2AX (Fig. 5A).
  8. Authors described “perilla callus extracts induced G1/S cell cycle arrest by reducing CDK6 levels, and that perilla extracts, particularly callus extracts, effectively regulate cell cycle-check point proteins and trigger DNA repair response in UVB exposed keratinocytes” (line 334). Authors should examine the effect of perilla leaf extracts on cell cycle arrest, levels of CDK6 and cell cycle check point proteins and DNA repair in mouse skin after UVB exposure. The reviewer suggests that the analysis would be performed at earlier time points than 2weeks after UVB exposure.
  9. Please correct “colony formation in after being” to “colony formation after being” in line 17.
  10. Please correct “CHK6” to “CDK6” in line 19.
  11. Please correct “Figure 4B-C” to “Figure 4A” in line 227.
  12. Please correct “Figure 6A” to “Figure 6” in line 299.

Reviewer 2 Report

Manuscript of Lee and Park describe the preparation of a plant extract from Perilla frutescens, and its effects in preventing DNA damage induced by UVB exposure.

The study is well designed, the introduction provides sufficient background, the manuscript is well organized, and I believe it is interesting for the readers of Nutrients.

However, I have several concerns that are listed in the following paragraphs:

  • The name of the species must be in italic. That does not happen throughout the manuscript. Please correct.
  • Material and Methods section 2.2 is confusing. What is mops? Is it an abbreviation? If so, please provide the complete name. Also, please remove the expression “as namely”. It does not make sense in that context.
  • Lines 198-202 - the sentences presented in this lines are confusing and it is not clear what the authors want to state. Please clarify.
  • In figure 5D, where it reads “Cell cycle phage (%)” I believe there is a typo and it should read “Cell cycle phase (%)”. Please correct
  • Lines 279-282 – “suggest that UVB-induced cell cycle arrest at G1/S could be effective in restoring the cell cycle and protecting the keratinocytes, which were inhibited by the perilla callus extracts.” These sentence is not clear. What is inhibited by the callus extracts? The cell cycle arrest? The protection of keratinocytes? Please clarify.
  • Lines 284-290 - the sentences presented in this lines are confusing and it is not clear what the authors want to state. Please clarify
  • Lines 327-328 - I do not understand what the authors mean to say with “ (…) the effect of perilla in ROS induced DNA damage in associated with cell cycle profiling (…)” Please clarify.
  • Lines 336-337 – “According to the data, perilla may play a role in potential natural compounds following to UV induced DNA damage response.” This sentence is not clear. Please rewrite to clarify.
  • Lines 339-342 - there are no references to support the affirmation made. Please add supporting references.
  • The final list of references is not formatted according to the instructions from Nutrients.

Also, according to the Journal’s Ethical Guidelines for the Use of Animals in Research:

“Manuscripts containing original research on animal subjects must have been approved by an ethical review committee. The project identification code, date of approval and name of the ethics committee or institutional review board must be cited in the Methods section.”

That information is not found in the manuscript. Please add that information

Reviewer 3 Report

The study “Perilla frutescens extracts enhance DNA repair 2 response in UVB damaged skin” by Hyuna Lee and Eunmi is interesting. However, there are several issues that need to be corrected. Many grammatical issues were seen as well as the flow of language and clarity needs to be improved. It seems that after the arrangement of the manuscript in the template provided by the journal authors have not proofread it. There are many mistakes in text formatting such as missing proper use of italic text, superscripts and subscripts, etc. Hence authors should carefully proofread the entire manuscript to omit these careless mistakes. The lack of essential information such as the chemical composition of the extract is a major drawback.

Please correct the following issues

Title – “Perilla frutescens extracts enhance DNA repair 2 response in UVB damaged skin” is a scientific name so it should be written as such (capitalization of the genus name and use of italics). Correct this mistake throughout the text.

Line 11 – Again, “perilla frutescens” is a scientific name so it should be written as such.

Line 10 – “Here, we investigated whether the leaf and callus of 10 perilla frutescens, as well known Asian herb, extracts affect DNA damage response and repair in 11 skin and keratinocytes exposed to UVB light.” The sentence needs to be revised. For example “Here, we investigated whether the leaf and callus extracts of Perilla frutescens, a well-known Asian herb, affect DNA damage response and repair in skin and keratinocytes exposed to UVB light.”

Line 15 – “30mJ/cm2” A “space” should be placed between the number and the unit. Correct this mistake throughout the text. In some places, it is written correctly.

Line 26- Categorizing “oxidative stress” as an intrinsic factor is incorrect. The effects of UV exposure proceed via increasing cellular oxidative stress. The authors need to correct this sentence.

The introduction should be structured more effectively and strengthened by incorporating relevant information regarding the mechanism by which UV exposure would cause the detrimental effects of “skin aging”. UV exposure would indeed cause DNA damage and it plays a part in skin aging. However, the symptoms of skin aging cannot only be explained by “DNA damage”. Start with a brief explanation of what are the few most common symptoms of UV exposure.

Line 35 – Scientific name

Line 37 – 42 authors have to improve the writing. The flow of language is poor.

Did the animal experiments were carried out according to institutional guidelines? Need to mention that in the manuscript

Line 60- “Perilla frutescens were collected in the Geumsan…” should be corrected as “Perilla frutescens were collected from Geumsan…”

From where did the authors purchase the HaCaT cell line?

Line 84 – “CO2” the no “2” should be subscripted. Correct this mistake throughout the text.

Line 87 – Normally, most of the researchers use serum-free media after UV exposure. So, please mention if the authors did the same.

Line 91 – “Cells were plated at 5×105 cells/well…” The power should be superscripted. Correct this mistake throughout the text.

The text format of the sign “°C” in line 107 and 109 are different.

Line 108 When writing chemical formula “Na2EDTA” please follow the general rules of subscripting the valence number.

A separate subsection “2.1. Materials” under the section “2. Materials and Methods” should be created. It should list the purchasing information of chemicals and reagents.

Line 100 – During the 14 days of incubation, didn’t the authors change the culture media? I wonder how the cells would survive for that long without the replacement of the media.

Line 102 – please provide a suitable reference for the “comet assay”.

I doubt if it is possible to observe a comet tail after just 3 h of UVB exposure. Usually, in HaCaT cells it takes 4-5 hours even for the observation of mitochondrial depolarization, which causes the initiation of the mitochondria-mediated apoptosis pathway. It is after that (6-7 hours) the DNA starts to get cleaved. So in order to get a clear comet tail, the cells should be harvested at least after 12 h from the UVB exposure. 24 h is the most ideal whereas you could observe a clear variation of the comet tail.

Line 117 didn’t the authors used RNase during cell cycle analysis?

Line 155 – Do the authors report the outcome of a single experimental trial or have them repeated the experiments to calculate the repeatability of the results? How many trials were carried out and in each trial how many replicates were dome? The authors should mention this information in the Figure legend.

Line 179 – Same comment as before

Figure 3 - I can’t see any comet tail even in the UVB+ group. I suggest authors to do the comet assay at least 12 h after the UVB exposure as 3 h is not enough to observe such effect in HaCaT cells.

Line 298 – “…reduced levels of cyclin CDK6 in…” should be “…reduced levels of cyclin D1 and  CDK6 in…”

Didn’t the authors check the chemical composition of the extract using basic compositional analysis assays? (I’m not mentioning about identifying active compounds which would be far better.) This is a major drawback of this study.

In the discussion mention previous studies that report the identification of bioactive compounds from Perilla frutescens (if there are any).